

# Induced density correlations in a sonic black hole condensate

Yi-Hsieh Wang[1,2], Ted Jacobson[3], Mark Edwards[1,4] and Charles W. Clark[1,2]

**1** Joint Quantum Institute, National Institute of Standards and Technology and
the University of Maryland, College Park, Maryland 20742
**2** Chemical Physics Program, University of Maryland, College Park, Maryland 20742
**3** Department of Physics and Maryland Center for Fundamental Physics,
University of Maryland, College Park, Maryland, 20742
**4** Physics Department, Georgia Southern University, Statesboro, GA 30460

## Abstract

Analog black/white hole pairs, consisting of a region of supersonic flow, have been achieved in a recent experiment by J. Steinhauer using an elongated Bose-Einstein condensate. A growing standing density wave, and a checkerboard feature in the density-density correlation function, were observed in the supersonic region. We model the density-density correlation function, taking into account both quantum fluctuations and the shot-to-shot variation of atom number normally present in ultracold-atom experiments. We find that quantum fluctuations alone produce some, but not all, of the features of the correlation function, whereas atom-number fluctuation alone can produce all the observed features, and agreement is best when both are included. In both cases, the density-density correlation is not intrinsic to the fluctuations, but rather is induced by modulation of the standing wave caused by the fluctuations.


## 1  Introduction

Unruh proposed in 1980 [1] that Hawking radiation could be observed in a sonic analog of a black hole. The black hole spacetime in this analogy is formed by a stationary fluid flow from a subsonic region to a supersonic one. The analog Hawking quanta are phonons, and the equivalent energy of the Hawking temperature $T_H$ is $\hbar$ times the gradient of the phonon velocity, evaluated at the sonic horizon, where the phonon velocity relative to the horizon vanishes.

Bose-Einstein condensates (BECs) provide a promising candidate for a black hole analogs of this type [2]. Indeed, observations of Hawking radiation in two experiments using a dilute, cigar shaped BEC of $^{87}$Rb have recently been reported [3,4]. The black hole horizon (BH) is created in these experiments by sweeping a step potential across the initially static, trapped condensate. The inside of the BH lies downstream from the step, where the flow is supersonic with respect to the reference frame of the step. The sound speed and healing length in these condensates are of order 1 mm/s and 1 $\mu$m respectively, so the velocity gradient can be of order 1/ms, corresponding to a Hawking temperature of order 10 nK. The BEC temperature is much lower than this, making possible the observation of spontaneous Hawking radiation, as reported in [4].

The directly measured observable in these experiments is the local density of atoms $n(x)$, which is probed by phase contrast imaging with micron resolution. From this one can measure the (connected) density correlation function,

$$
\begin{aligned}
G^{(2)}(x, x') &= \langle n(x)n(x')\rangle - \langle n(x)\rangle\langle n(x')\rangle \\
&= \langle \delta n(x)\delta n(x')\rangle,
\end{aligned}
\tag{1}
$$

where $\delta n(x) = n(x) - \langle n(x)\rangle$ is the fluctuation away from the mean value. Expectation values can, in principle, be evaluated by repeating the measurement many times, with the condensate prepared in the same initial state each time. With the correlation function and its Fourier transform, the spatiotemporal structure and spectra of the phonon excitations can be robustly measured [5].

In practice, the measured correlation function is not the quantum expectation value, because it inevitably includes an average over any other aspects of the experiment that vary. While the trap and step potentials are controlled with high precision, in typical BEC experiments the number of atoms in the trap varies from run to run at levels much larger than shot noise. In this paper we show that this variation can have important consequences for the correlation function, affecting its physical interpretation.

In the experiment of Ref. [3], a key role is played by the inner, or "white hole" horizon (WH), which lies further downstream where the trap potential slows the atoms to subsonic velocities. There is an effective "cavity" between the horizons, which traps the negative energy partners of Hawking radiation. In the experiment, and in simulations using the Gross-Pitaevskii (GP) equation for the mean field condensate wavefunction, it has been found that

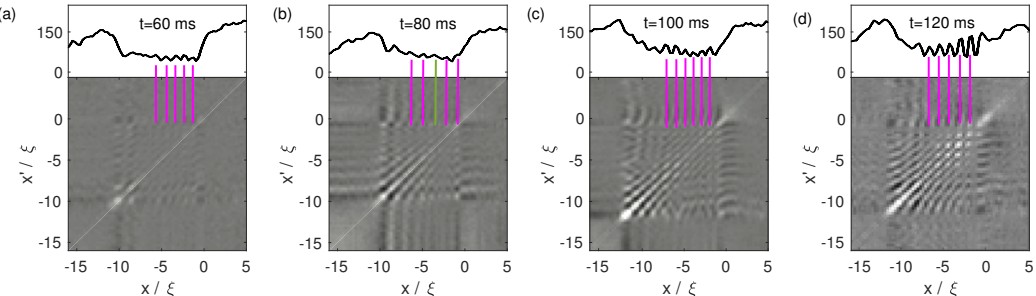

Figure 1: Experimental density and density-density correlations taken from [3]. Top: the ensemble average of experimental density, $\langle n_{\exp} \rangle$; bottom: density-density correlation. Panels (a)-(d) are measured at times $t = 60, 80, 100, 120$ ms, respectively. The periodic features of the standing waves are very close to those of the checkerboard patterns in the correlation, as indicated by the magenta vertical lines; the green line in panel (b) indicates the only mismatch, where the standing wave peak falls in a trough of the correlation pattern. Note that all the plots are based on the length unit $\xi$ of the system, $\xi = 2$ $\mu$m, and the origin $x = x' = 0$ is defined at the step edge (i.e. the BH).

a zero frequency standing wave arises in the cavity, extending from the WH to the BH [6–8], as shown in Fig. 1(a-d) and Fig. 2(a). For a single solution of the GP equation, the correlation function vanishes, but nonvanishing correlation functions displaying a checkerboard feature were found in Refs. [6,8], when including random fluctuations around the initial mean field wave function [6,8] or in the height of the step potential [6]. The latter affects the location of the WH horizon.

We simulate the density correlation with quantum fluctuations (using the truncated Wigner method [9–13]) and also observe a checkerboard feature in the supersonic region. However, we find much better agreement with the experimentally measured correlation function when including variations in the atom number along with quantum fluctuations. In fact, the checkerboard and other features in the correlation function are quite well matched when *only* atom number variations are included. That both types of fluctuations produce similar correlation functions is a consequence of the presence of the large standing wave. The standing wave is modulated by the variation of atom number, and by the long-wavelength components of quantum fluctuations. This induces a nonvanishing correlation displaying a checkerboard pattern near the BH horizon, as well as (particularly for number variations) diagonal streaks near the WH horizon. With both quantum fluctuations and atom number variations included, the overall correlation function is in good qualitative agreement with the observed one in all respects.

## 2 Methods

In the experiment, the condensate dynamics was nearly one dimensional [3], and we therefore use the one dimensional Gross-Pitaevskii equation (1DGPE) to model the condensate. For the trap potential, we used the form of a Gaussian beam potential along the axis of symmetry, choosing parameters that optimize the fit to the experiment. Details are given in Appendix A.

To simulate the variation of atom number $N$ from shot to shot under experimental conditions, we calculate the condensate properties using a normal distribution of $N$ with mean $\bar{N} = 6000$, and standard deviation $\Delta N$. We consider three different values, $\Delta N = (0.05, 0.1, 0.15)\bar{N}$. (Note that the shot-noise number variation is given by

$\sqrt{N} \sim 0.013N$.) For each value of $\Delta N$, we ran 200 simulations of the experiment reported in [3], each with a random choice of $N$, and computed the average density and the density correlation function at one particular time.

To simulate the effects of quantum fluctuations in the condensate, we use the truncated Wigner approximation (TWA) [9–13]. In this method, the linearized perturbations of the GP wave function for the initial, stationary condensate, i.e. the Bogoliubov-de Gennes (BdG) modes, are populated with random phases, and with amplitudes according to the probability distribution defined by their zero-temperature quantum state. Because of the adiabatic theorem, the modes with frequencies much higher than those that are dynamically relevant in the system should not affect the evolution. In our system, the dynamics of interest are: (i) the black hole lasing effect, which is bounded by the maximal frequency $\omega_{\max}$ in the dispersion relation of the supersonic flow, as mentioned in [3], (ii) the background standing wave, which has a nonzero frequency in the BH frame but much smaller than $\omega_{\max}$ [7]. We limit the number of BdG modes to $K = 200$, so that the frequency of the last mode is much greater than $\omega_{\max}$. We also test the simulation by increasing the number of modes, and the resulting correlation does not change much.

When investigating the effect of the quantum fluctuations alone, we adjust the amplitude of the unperturbed part of the GP wave function for each realization so that the total $N$, after including the fluctuations, is the same for all realizations. Details are given in Appendix B.

## 3 Results

### 3.1 Experimental density-density correlation

In the experiment of [3], a BH/WH cavity was generated by sweeping a step potential through an initially stationary condensate, and the density-density correlation function at a given time was measured by repeating the experiment 80 runs for each time. The correlation function defined in Ref. [3] includes, in addition to Eq. (1), the term $-\langle n(x)\rangle \delta(x-x')$. This extra term has support only on the diagonal, and is singular there. Ref. [3] does not indicate how this term was handled, and here we do not include it.

The top panels of Figs. 1(a-d) [3] show the evolution of experimentally measured ensemble averages of the individual density profiles, featuring a growing standing wave behind the BH horizon (at $x = 0$), and the bottom panels are the corresponding correlation functions, which feature a square array pattern in the upper right quadrant. This was called a "checkerboard" pattern in [3]. We compare the standing wave with the checkerboard near the BH, for $t \geq 60$ ms, after both features have developed and can be observed clearly. The periodic features of the correlations match those of the density profiles very well, as indicated by the magenta lines, strongly suggesting that the two features have the same origin. Note that there is one mismatch, indicated by the green line in Figs. 1(b), where the standing wave peak falls in a trough of the correlation pattern. In the following, we investigate the roles of atom-number variation and quantum fluctuations in producing the correlation, and analyze how the standing wave is related to the observed correlation function.

### 3.2 Simulations with atom number fluctuations and quantum fluctuations

Figure 2(a) displays the result of a single GP simulation at $t = 120$ ms, with a definite atom number and no quantum noise. A standing wave is seen inside the supersonic cavity, extending from the BH horizon at $x = 0$ to the WH horizon at $x \sim -10\xi$ ($\xi$ denotes the healing length in the cavity region, estimated in the experiment to be $\xi = 2$ $\mu$m [3]). The Fourier transform of $n(x)$ over a square window behind the BH horizon ($-5.6\xi < x, x' < -1.8\xi$, similar to the

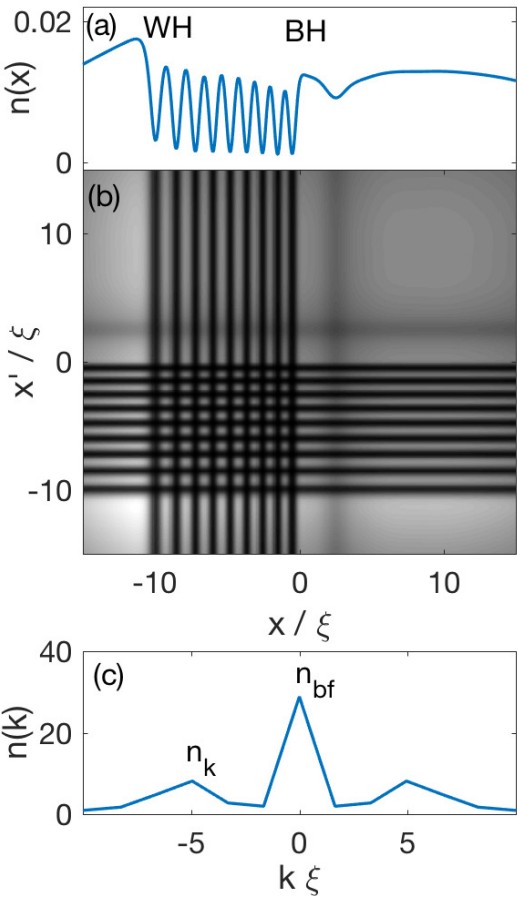

Figure 2: GP simulation and its wavevector spectrum at $t = 120$ ms. (a) density plot $n(x)$; (b) density-density plot, $n(x)n(x')$; (c) wavevector spectrum $n(k)$ for the condensate density between the BH and the WH. The central peak indicates the background density, $n_{\text{bf}}$, and the side peaks indicate the standing wave amplitude, $n_k$.

window in Fig. 4(a) of [8]) gives rise to the wavevector spectrum shown in Fig. 2(c), with two side peaks corresponding to the standing wave, $n_k$, and a central peak coming from the background flow, $n_{\text{bf}}$. The correlation function vanishes identically for a single deterministic simulation, but we show in Fig. 2(b) the density-density function $n(x)n(x')$, for the purpose of comparison with what is to come.

Figure 3 shows the results of simulations incorporating atom-number fluctuations (NF), quantum fluctuations (QF), and both types of fluctuations, as well as the experimental results. The bottom panels show the correlation functions. The top panels show the ensemble average of density $\langle n(x) \rangle$ (black curve) and, in (e-g), a single random realization of the simulations, $n_i(x)$ (red curve). Figure 4 shows the 2D wavevector spectra of the correlation functions in Fig. 3, performed over the quadrant $-5.6\xi < x, x' < -1.8\xi$ in (a-c), and the top panels show the cut-through at $k'\xi = -5$. Panel (d) shows the experimental wavevector spectrum, calculated using the digitized correlation data in Fig. 3(d), originally displayed in Fig. 4(b) of [8]. The correlation functions are shifted by a constant to remove the peak at $k = k' = 0$, as in Fig. 4(b) of [8] (the spectra are pixelated due to the window we adopted to match Fig. 4(a) of [8]. The resolution could be improved by the method of windowed Fourier transform, as adopted in [7, 14]).

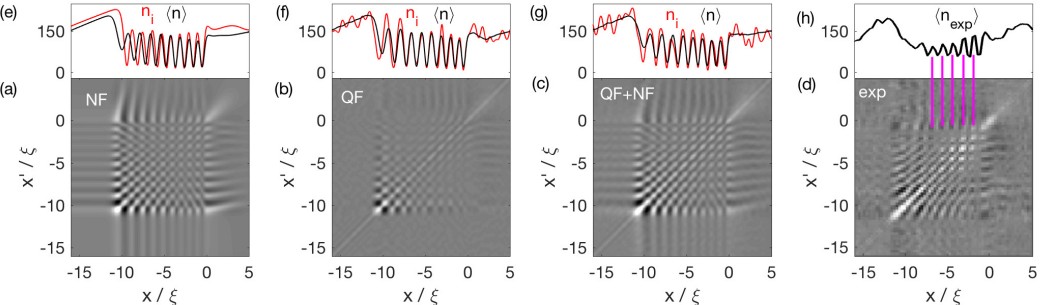

Figure 3: The density-density correlations at $t = 120$ ms by (a) number fluctuations, (b) quantum fluctuations, and (c) both. (d) experimental density-density correlation taken from [3]. Note that for panels (a) and (c), the number of condensate atoms fluctuates about $\Delta N / \bar{N} = 0.05$. Top panels (e-g) are the profiles of the averaged density $\langle n(x) \rangle$ (black) and that of one realization in the corresponding ensemble , $n_i(x)$ (red). Panel (h) is the ensemble average of experimental density, $\langle n_{\text{exp}} \rangle$, taken from [3].

### 3.2.1   Atom number fluctuations

Figures 3(a,e) correspond to the case of fluctuating atom number $N$ ($\bar{N} = 6000$, $\Delta N = 0.05\bar{N}$), without quantum fluctuations. The correlation function contains a checkerboard similar to that in the experimental plot. Also, near the WH horizon, the pattern is partially smeared out into lines parallel to the diagonal. Figure 4(a) shows the 2D wavevector spectrum of Fig. 3(a) and its cut-through at $k'\xi = 5$. The peaks at $k\xi = \pm 5 \sim \pm 6$ are consistent with the spacing of the squares in the checkerboard ($\sim 1\xi$). The peak at $k = 0$ indicates a non-oscillatory component. The 2D Fourier transform is quite similar to that in the experimental plot, Fig. 3(d), the principal differences being that in the simulation the peaks are somewhat more sharply defined and do not differ as much in intensity.

### 3.2.2   Quantum fluctuations

Figures 3(b,f) show the result of including quantum fluctuations using TWA simulations at zero temperature, with a fixed total atom number. In each run, these fluctuations correspond to "noise" added at the beginning of the evolution, which evolves with the condensate, and whose effect on a random realization is shown in the red curve of the top panel. The correlation function also contains a checkerboard in the cavity region, but not as distinct as in the NF case near the BH horizon. The bright diagonal line is a feature resulting from the quantum noise, which adds up constructively at $x = x'$ [11]. The 2D Fourier spectrum for Fig. 3(b) and its cut-through are shown in Fig. 4(b). As in the previous case, we find peaks at $k\xi = \pm 5 \sim \pm 6$ corresponding to the checkerboard. In addition, there is an off-diagonal line, which arises from the diagonal line in the correlation function (since the Fourier transform of $\delta(x - x')$ is $\delta(k + k')$), and the off-diagonal peak is thus enhanced relative to the other peaks.

### 3.2.3   Atom number and quantum fluctuations

Figures 3(c,g) show the result of incorporating both types of fluctuations together into the simulation. For each realization, we randomly select a condensate atom number $N_i$ to determine the initial condensate, then introduce quantum fluctuations on top of this, before proceeding with the simulation. (The addition of quantum fluctuations only increases the total atom number by a small amount, ($\approx 0.006N_i$), which is much smaller than atom-number fluctuations

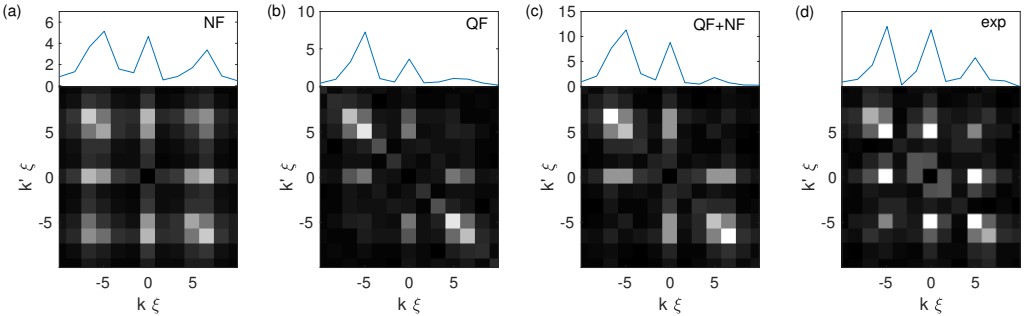

Figure 4: 2D wavevector spectra for the correlations in Fig. 3 by (a) number fluctuations, (b) quantum fluctuations, and (c) both. Bottom: 2D wavevector spectrum; top: cut-through of the 2D spectrum at $k'\xi = 5$ (note different scales on the plots). Panel (d): the experimental wavevector spectrum calculated from the digitized correlation data in Fig. 3(d); upper panel is its cut-through at $k'\xi = 5$. Note that for panels (a) and (c), the number of condensate atoms fluctuates about $\Delta N/\bar{N} = 0.05$. For all the 2D spectra, the value at $k = k' = 0$ is calibrated to zero, which corresponds to a constant shift of the correlation function.

we consider here.) The checkerboard near the BH is consistent with the pure NF case, and the presence of the diagonal line is due to the quantum fluctuations. The correlation function closely resembles the experimental one, Fig. 3(d,h), regarding the checkerboard near the BH, the diagonal line, and the parallel lines near the WH. The corresponding 2D wavevector spectrum is shown in Fig. 4(c). Compared with the pure NF case (Fig. 4(a)), the off-diagonal peaks are enhanced by the addition of quantum fluctuations, and the diagonal ones are suppressed. This is qualitatively similar to the experimental spectrum Fig. 4(d).

### 3.2.4 Comparison of different atom number variances

Figures 5(a-c) display the results incorporating both quantum fluctuation and number fluctuation, with three different values for the standard deviation, $\Delta N = (0.05, 0.1, 0.15)\bar{N}$. The top panels (e-g) show the ensemble average of density, $\langle n(x)\rangle$, given by the black curve, and that of a single, random realization, $n_i(x)$, given by the red curve. Figures 6(a-c) show the 2D wavevector spectra of Figs. 5(a-c). For comparison, we show the experimental results on the rightmost panel of Figs. 5, 6.

For all three number variances, the structure of the density variations near the BH horizon, the checkerboard patterns in the right half of the cavity, and their spectra in simulation and experiment agree fairly well with the experiment. The smeared lines parallel to the diagonal in the correlation function match the experiment better in the $\Delta N/\bar{N} = 0.1, 0.15$ cases. The overall experimental density profile $n(x)$ decreases sharply from the BH horizon to the WH horizon, a behavior that is best matched for $\Delta N = 0.15\bar{N}$.

### 3.3 Influence of atom number and quantum fluctuations on standing wave and correlation

In this subsection we suggest some mechanisms for the effect of number and quantum fluctuations on the standing wave and correlation function, and we briefly consider also some other simulations described in the literature. As a preliminary comment, we found that very small modifications of the strength of the trap potential can have a relatively large effect on certain details in the evolution of the condensate. This sensitivity to the potential is demonstrated in Appendix A, where it is shown that a 3% variation in the overall coefficient of the

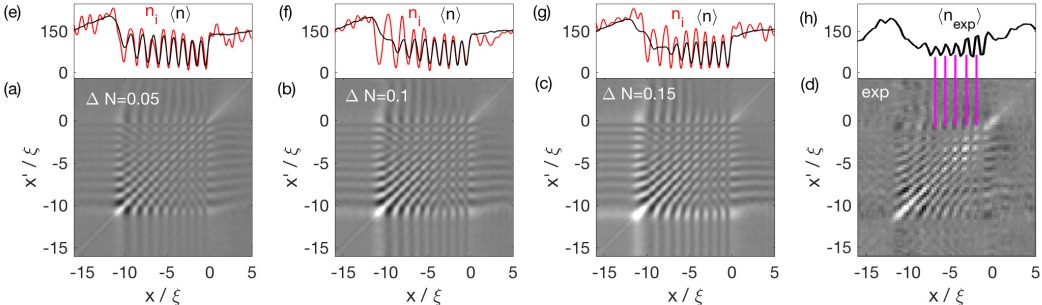

Figure 5: The density-density correlations at $t = 120$ ms by both number and quantum fluctuations. For panels (a-c), the number of condensate atoms fluctuates about $\Delta N/\bar{N} = 0.05, 0.1, 0.15$, respectively. Panel (d): experimental density-density correlation taken from [3]. Top panels (e-g) are the profiles of the averaged density $\langle n(x) \rangle$ (black) and that of one realization in the corresponding ensemble , $n_i(x)$ (red). Panel (h) is the ensemble average of experimental density, $\langle n_{\exp} \rangle$, taken from [3].

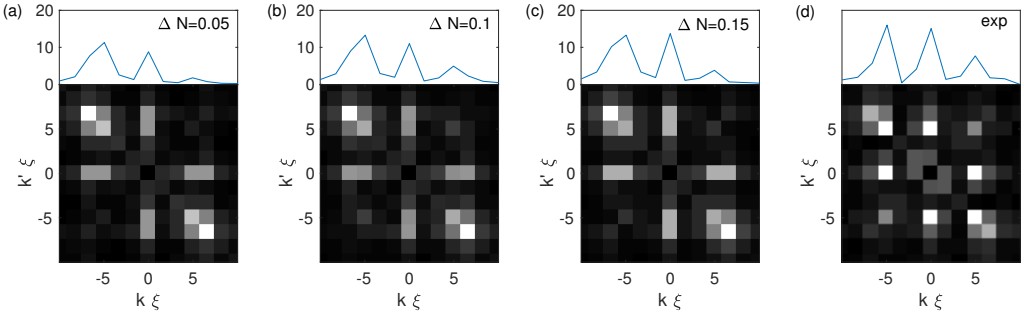

Figure 6: 2D wavevector spectra for the correlations in Figs. 5. Panels (a-c): $\Delta N/\bar{N} = 0.05, 0.1, 0.15$, respectively. Bottom: 2D wavevector spectrum; top: cut-through of the 2D spectrum at $k'\xi = 5$. Panel (d): 2D wavevector spectrum of the experimental correlation; upper panel is its cut-through at $k'\xi = 5$.

(one-dimensional) trap leads to measurable differences.

Atom number fluctuations influence both the amplitude of the background flow, and the amplitude and phase of the cavity standing wave. Figure 7(a) shows the density variation, $\delta n = n - \langle n \rangle$ (where $\langle n \rangle$ is the average density for the ensemble with standard deviation $\Delta N = 0.05\bar{N}$) as a function of position and atom number variation $\delta N$. The correlation function is the product of the density variations $\delta n(x)\delta n(x')$, averaged over the different number values in the normal distribution, with 68% weight from $|\delta N|/\bar{N} < 0.05$, and another 27% weight from $0.05 < |\delta N|/\bar{N} < 0.1$, and only 5% from outside the region of the plot. The phase varies with $N$ more in the left half of the cavity, closer to the WH, but also varies on the right half. However in the right half of the cavity, near the BH, the response to number fluctuations is markedly weaker, and asymmetric, being stronger for negative fluctuations than for positive ones. Therefore in that region the correlation function is more influenced by negative number fluctuations.

The plot in Fig. 7(b) shows the average density (dashed black curve, $\langle n \rangle$) together with two realizations from the ensemble: one with $\delta N = 0.075\bar{N}$ (red curve, $n_{\max}$), and one with $\delta N = -0.075\bar{N}$ (blue curve, $n_{\min}$). Compared to $\langle n \rangle$, the amplitudes of $n_{\max}$ and $n_{\min}$ are above and below, respectively, and for $n_{\max}$ the left edge of the cavity shifts towards the right,

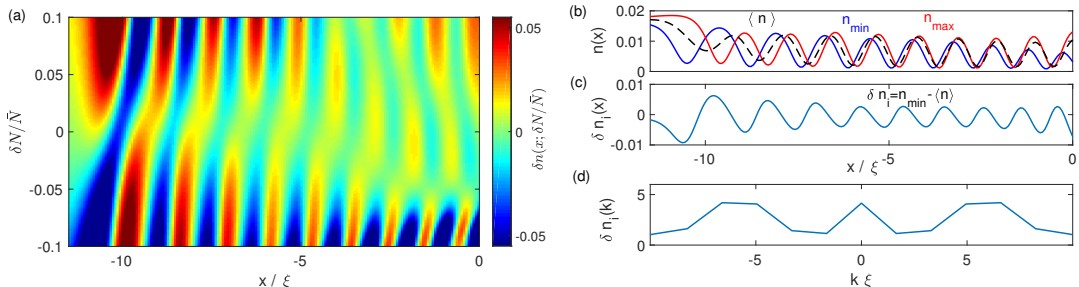

Figure 7: Effects of atom-number variation on the standing wave. (a) density variation, $\delta n = n - \langle n \rangle$, as a function of position, for atom number $N = \bar{N} + \delta N$. (b) dashed black: averaged density over atom-number fluctuation, $\Delta N / \bar{N} = 0.05$; solid red: density of a realization with atom number $N = \bar{N} + 0.075\bar{N}$, $n_{\max}$; solid blue: one with atom number $N = \bar{N} - 0.075\bar{N}$, $n_{\min}$. (c) difference between the density with $N_{\min}$ and the averaged density, $\delta n = n_{\min} - \langle n \rangle$. (d) wavevector spectrum for $\delta n(x)$ in a region near the BH, $-5.6\xi < x < -1.8\xi$.

while for $n_{\min}$ it shifts towards the left. The density variation $\delta n = n_{\min} - \langle n \rangle$ is shown in Fig. 7(c). It has a wavelength similar to that of the density itself, and its wavevector spectrum, shown in Fig. 7(d), has peaks close to those of the average density spectrum shown in Fig. 2(c). The contribution from zero wavenumber corresponds to the overall modulation of the total number of atoms, which can be seen in the single realizations in position space shown in Fig. 7(b). The variation $\delta N$ thus modulates the overall condensate density, and it also changes the location of the WH horizon, and thus modulates the phase of the standing wave at a given location.

Atom number fluctuations therefore result in a fluctuating $\delta n$, with a characteristic wavenumber equal to that of the standing wave, as well as a background ($k = 0$) component, and hence a nonzero correlation function $\langle \delta n(x) \delta n(x') \rangle$ with that same wavenumber. The pattern that emerges is different in the right and left halves of the cavity, because of the differences in the intensity of the variation and the shifting of the phase of the oscillation. On the left half of the cavity, the phase fluctuates more, hence the correlation function is smeared out to be relatively constant on diagonal lines of constant $x' - x$, as seen in the correlation function with number fluctuations, Fig. 3(a), and observed in the experiment Fig. 3(d). On the right half, there is little density variation except for the larger negative number variations, so a fairly constant phase contributes, and the resulting correlation function is therefore similar to the plot of $n(x)n(x')$ shown in Fig. 2(b), which is similar to the checkerboard pattern seen in the figures just mentioned. Note that this "checkerboard" differs from an ordinary checkerboard pattern, in that there are wide dark nodes, rather than an alternating pattern of adjacent light and dark squares. The $k = 0$ component of $\delta n$ is essential for the occurrence of these dark nodes. Without it, the correlation function would be something like $\cos kx \cos kx'$, whereas with it the correlation function is more like $(A + \cos kx)(A + \cos kx')$, where $A$ is a constant.

Quantum fluctuations alone also produce a correlation function. Figure 3(f) shows the background standing wave in the cavity, with the addition of fluctuating spatial noise. The standing-wave amplitude for each run varies slightly from the average profile, while the phase is less affected than for number fluctuations, since the zero-point field does not change the overall flow structure. Such variation results in a faint checkerboard, seen in Fig. 3(b), and its wavevector spectrum in Fig. 4(b) shows peaks at the same values as produced by number fluctuations but weaker, and with an off-diagonal ($k' = -k$) contribution much larger than the other contributions. The checkerboard feature of this correlation function, including the dark nodal lines, might arise as follows: the GP wavefunction has the form $\Psi(x, t) = \Psi_0(x, t) + \delta\psi(x, t)$, where $\Psi_0(x, t)$ is the wavefunction without the quantum noise

having been added at $t = 0$. The dominant contribution to the density-density correlation function will come from the cross terms in the density $n(x) = \Psi^{\dagger}(x)\Psi(x)$ between $\Psi_0$ and $\delta\psi$. The components of $\delta\psi$ with wavelength long compared with that of the standing wave thus modulate the amplitude of the background and standing wave in $\Psi_0$, in a spatially correlated fashion. We check this interpretation by simulating the quantum noise from only the short wavelength modes, and the resulting correlation does not have the checkerboard pattern.

### 3.3.1   Comparison with other simulated results

Correlation functions similar to Fig. 3 are simulated and reported in Refs. [6,8]. These simulations include noise as a proxy for the effect of quantum fluctuations (local Gaussian noise in the case of [6], and noise induced by a short pulse Bragg technique in the case of [8]). The effect of fluctuations in the WH horizon location induced by fluctuations in the step potential were also explored in [6].

Figure 5(a) of Ref. [6] and Fig. 4(c) of Ref. [8] show a similar checkerboard pattern, which alternates between black and white in the region near the BH. This is somewhat consistent with our TWA simulation, regarding the black-white-alternating feature; however, for our QF simulations, the dark nodes (discussed above in this subsection), while not as distinct as in the NF case, are more evident. This difference is also manifested in the Fourier transform, which has more power at $k = 0$ in our simulation.

Figure 5(b) of Ref. [6] shows the correlation function for the ensemble with fluctuations in the step potential, and therefore fluctuations in the position of the WH horizon. This correlation function displays diagonal streaks, consistent with what one expects when the phase of the standing wave is fluctuating within the ensemble, across the entire cavity between the WH and BH horizons. This is similar to the pattern we have seen induced by number fluctuations on the WH side of the cavity, but not on the BH side. It should also be noted that the step potential fluctuations introduced in [6] were more than two orders of magnitude larger than in the experiment [8], whereas the number fluctuations we have introduced are comparable to those encountered in BEC experiments. Ref. [6] also mentions finding that atom number fluctuations have no significant effect on observables, and in particular do not change the position of the WH horizon. As discussed in Appendix A, we found that the condensate is more affected by number fluctuations for a shallower trap. This might explain why Ref. [6] found no significant effect of number fluctuations, as the trap potential shown in their Fig. 1 appears to be steeper than the one we used.

In summary, while the noise added in the simulations of Refs. [6,8] does have the effect of eliciting correlation functions that reflect the structure of the background standing wave, the correlation function we obtain combining number fluctuations and quantum fluctuations via the TWA is significantly closer to the experimentally observed one.

## 4   Discussion

We have found that atom number fluctuations play a dominant role in giving rise to density-density correlations in a BEC with a background density wave structure. This, together with a subdominant, similar contribution from quantum fluctuations, appears to account for all features of the checkerboard correlation pattern observed in the BH/WH cavity in the experiment of [3]. Although the details of the correlation function depend on the types of fluctuations, and on small variations of trap and step potentials, the main checkerboard pattern is a robust feature, which exhibits a high degree of similarity with the checkerboard in the density-density plot, Fig. 2(b). Likewise, the spectrum of the correlation function in each case is consistent with that of the background standing wave, Fig. 2(c). This indicates that quantum fluctuations,

and number fluctuations, only induce in the correlation function a pattern that is already established by the standing wave.

In Ref. [7] we modeled this experiment using the GP equation without any fluctuations, and found that the standing wave corresponds to zero-frequency Bogoliubov-Čerenkov radiation (BCR), originating at the WH horizon where the flow transitions from supersonic to subsonic. This frequency is Doppler shifted to a nonzero value in the reference frame of the BH horizon (because the WH horizon is receding from the BH horizon), where it stimulates Hawking radiation at that frequency. We found no sign of the black hole laser instability [15,16] that can in principle take place in this configuration, and inferred that the observed phenomena are driven by the BCR alone. However, since our previous analysis did not include any fluctuations, it was unable to produce the observed correlation function, and was unable to demonstrate explicitly that the addition of quantum fluctuations does not trigger the laser instability (although the condensate in the cavity is sufficiently inhomogeneous and time dependent to expect that if the instability could occur on the timescale of the experiment, it would have manifested in our previous GP simulations).

In this paper, we find that zero temperature quantum fluctuations, introduced using the TWA approach, do not seed a laser instability on the time scale of the experiment. This can be seen from the density profiles in Fig. 3(f), and from the comparison between the GP simulation and one TWA realization, shown in Fig. 10 in Appendix B. In the TWA realization, the standing wave amplitude matches closely with that of the GP simulation, and is only modulated slightly by the spatial noise of the QF. With and without quantum fluctuations, the only growing mode observed in the supersonic region is the BCR standing wave. The self-amplifying lasing mode is not observed in our simulation.

Finally, the possibility that the correlation function observed in other BEC experiments may be strongly affected by number fluctuations deserves to be investigated. In the future, for an experiment where it is important to suppress atom number fluctuations, they might be reduced below the $1/\sqrt{N}$ shot noise level using recently developed experimental techniques [17].

## Acknowledgements

We are grateful to G. Campbell, I. Carusotto, S. Eckel, A. Kumar, R. Parentani, J. Steinhauer, and E. Tiesinga for helpful conversations. This material is based upon work supported by the U.S. National Science Foundation Physics Frontier Center at JQI and grants PHY–1407744, PHY–1004975 and PHY–0758111, and by the Army Research Office Atomtronics MURI.

## A   GP simulation

In this appendix, we describe the procedures we used to simulate the experiment. We use the time–dependent Gross–Pitaevskii (GP) equation to determine the condensate wavefunction $\Psi(\mathbf{r}, t)$,

$$i\hbar \frac{\partial \Psi(\mathbf{r}, t)}{\partial t} = \left( T(\mathbf{r}) + U(\mathbf{r}) + g_{3D} N |\Psi|^2 \right) \Psi(\mathbf{r}, t), \tag{2}$$

where $T(\mathbf{r})$ and $U(\mathbf{r})$ are the kinetic and potential energy operators, $N$ is the number of condensate atoms, $g_{3D} = 4\pi\hbar^2 a/m$ where $a$ is the $s$–wave scattering length, and $m$ is the mass of a condensate atom. For a cylindrically symmetric system, the potential depends only on the

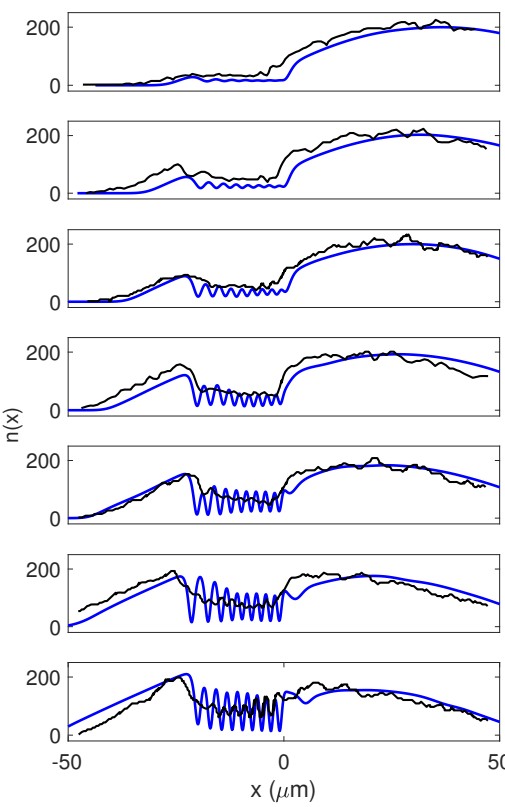

Figure 8: The evolution of a condensate. The density profile is plotted at 20 ms intervals with moving potential step $U_s = 0.75\mu$, scaled by a common factor to match experiment, and viewed in the moving frame where $x = 0$ defines the step edge. Black: experiment [3]; blue: present simulation.

radial coordinate $\rho$ and the axial coordinate $x$, $U(\mathbf{r}) = U(\rho, x)$; and $T(\mathbf{r})$ can be expressed as

$$T(\mathbf{r}) = -\frac{\hbar^2}{2m}\left(\frac{\partial^2}{\partial\rho^2} + \frac{1}{\rho}\frac{\partial}{\partial\rho} + \frac{\partial^2}{\partial x^2} + \frac{\partial^2}{\partial\phi^2}\right). \tag{3}$$

Regarding the ground state of the BEC, the azimuthal coordinate $\phi$ in the wavefunction can be suppressed, such that $\Psi(\mathbf{r}, t) = \Psi(\rho, x, t)$. Note that our definition of cylindrical coordinates is unconventional as regards the choice of x to define the azimuthal axis. This is done to maintain the consistency of our notation with that of Refs. [3, 4, 8].

For the experiment of interest [3], the trap potential is formed by a Gaussian laser beam:

$$U(\rho, x) = U_0\left[1 - \left(\frac{w_0}{w(x)}\right)^2\exp\left(\frac{-2\rho^2}{w^2(x)}\right)\right]. \tag{4}$$

where $U_0$ is the trap strength proportional to the peak laser intensity and

$$w(x) = w_0\sqrt{1 + \left(\frac{x}{x_0}\right)^2}, \qquad x_0 = \frac{\pi w_0^2}{\lambda}. \tag{5}$$

where $\lambda$ and $w_0$ denote the wavelength and the waist of the laser beam, respectively. According to [3], $\lambda = 812$ nm, $w_0 = 5$ mm, and the radial frequency is $\omega_\rho = 123$ Hz. This corresponds

to a trap tightly confined in the radial direction $\rho$, and elongated along the axial direction $x$. Expanding the trap at $x = \rho = 0$ to second order, we obtain an approximate harmonic potential:

$$U(\rho, x) \approx \left(\frac{2U_0}{w_0^2}\right)\rho^2 + \left(\frac{U_0}{x_0^2}\right)x^2 \equiv \frac{1}{2}m\omega_\rho^2\rho^2 + \frac{1}{2}m\omega_x^2 x^2. \tag{6}$$

from which the trap strength can be estimated by $U_0 = (1/4)m\omega_\rho^2 w_0^2 \approx k\,(39\,\text{nK})$, where $k$ is the Boltzmann constant.

When the system is tightly-confined in the radial direction, such that the integrated density $n(x)$ in the axial direction satisfies $an \ll 1$, it can be viewed as quasi-one-dimensional [18]. Were the potential cylindrically symmetric, one could approximate the wavefunction in the radial direction by the solution of a harmonic oscillator, such that

$$\Psi(\mathbf{r}, t) \approx \Phi(\rho)\Psi_{1D}(x, t), \tag{7}$$

where the radial wavefunction is $\Phi(\rho) = \exp\left[-\rho^2/\left(2d^2\right)\right]/(d\sqrt{\pi})$, $d = \sqrt{\hbar/\left(m\omega_\rho\right)}$, and $\Psi_{1D}(x, t)$ is the axial wavefunction. Integrating the GP equation in Eq. 2 over $\rho$ would then yield a 1D GP equation for $\Psi_{1D}(x, t)$, with an effective interaction coefficient $g_{1D} = g_{3D}m\omega_\rho/\hbar$:

$$i\hbar\frac{\partial\Psi_{1D}(x, t)}{\partial t} = \left(-\frac{\hbar^2}{2m}\frac{\partial^2}{\partial x^2} + U_{1D}(x) + \hbar\omega_\rho\right)\Psi_{1D}(x, t) + g_{1D}N\left|\Psi_{1D}(x, t)\right|^2\Psi_{1D}(x, t), \tag{8}$$

where $U_{1D}(x)$ is the axial trap.

However, the Gaussian beam potential (4) is not cylindrically symmetric. Rather than using a 2D potential, or attempting to incorporate the non-cylindrical effects in an improved approximation scheme, we elected to adopt a simple 1D potential capable of qualitatively matching the reported experimental results. (This is partially motivated by computational convenience, and partially by a lack of detailed knowledge of the experimental procedures by which the parameters characterizing the potential were measured.) Thus, for $U_{1D}(x)$ we use the Gaussian-beam potential (4) evaluated at $\rho = 0$, i.e. $U_{1D}(x) = U_0 x^2/(x^2 + x_0^2)$. Using this potential and 8, we simulate the step-sweeping experiment [3, 7]. We introduce a step potential $U_{\text{step}}(x, t)$, which takes the form

$$U_{\text{step}}(x, t) = -U_s(\tanh((x_s(t) - x)/D_s) - 1)/2. \tag{9}$$

Here $U_s$ is the step strength, $U_s = 0.75\mu$, with $\mu$ the chemical potential, and $D_s$ is the step width, $D_s = 0.5\mu\text{m}$. The position of the step is defined by $x_s(t) = x_0 + v_s t$, where $v_s = 0.21$ mm/s is the step speed, and $x_0$ is the position at $t = 0$.

We adjust the parameter $U_0$ from the estimate in Eq. 6, to optimize the consistency of the simulation with the experimental observations [3]. Figure 8 shows the evolution of the condensate density profile with $U_0 = k\,(33.2\,\text{nK})$. The single evolution of the 1D condensate agrees qualitatively well with the average density measured in the experiment, regarding the shape of the background condensate, the cavity size, and the wavelength and phase of the standing wave near the BH. As discussed in the text, the addition of atom number fluctuations to the simulated ensemble suppresses the oscillation amplitude on the left half of the cavity, and so improves the agreement.

We noticed that certain features of the condensate evolution can be very sensitive to parameters in the potential. To illustrate this here, we vary the trap coefficient slightly, by 3%, $U_0' = (1 \pm 0.03)U_0$, which is illustrated in Fig. 9(a). The resulting density profiles at $t = 120$ ms are shown in Fig. 9(b). The cavity size in the density profiles is similar, however there is a change in the standing wave pattern, and the density to the right of the cavity (where

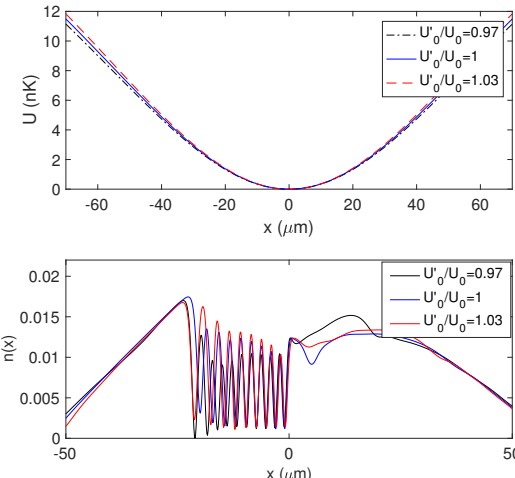

Figure 9: Panel (a): effective axial potential, $U_{1D}(x)$. Solid blue: $U_0'/U_0 = 1$; dashed-dotted black: $U_0'/U_0 = 0.97$; dashed red $U_0'/U_0 = 1.03$. Note that the minimum of $U(x)$ is shifted to zero. Panel (b): density profiles using the corrected potentials with parameter $U_0'/U_0 = 0.97$, 1, 1.03. Note that $x = 0$ defines the step edge.

the Hawking radiation is emitted), as shown in Fig. 9(b). We also found that this phase shift modifies the checkerboard pattern in the correlation function. The checkerboard is present in all cases, but the variation $\delta n(x)$ is more sensitive to the change of atom number with a lower trap. This produces stronger features of the lines parallel to the diagonal near the WH, and the dark nodal lines near the BH in the checkerboard. We conjecture that this hypersensitivity to the potential strength is related to the Čerenkov instability that produces the standing wave, since small differences in the condensate and flow early in the evolution may be amplified by the onset of the instability.

## B  Truncated Wigner method

In the TWA method [10, 12] adopted in the simulation, one includes a fluctuation term in the GP field, $\delta\psi(x)$, which models small excitations on a given stationary zero-temperature condensate, $\Psi_0(x)$:

$$\delta\psi(x,t) = \sum_j e^{-i\mu t}\left(\beta_j u_j(x)e^{-i\omega_j t} + \beta_j^* v_j^*(x)e^{i\omega_j t}\right). \tag{10}$$

Here the functions $u_j$ and $v_j$ satisfy the coupled Bogoliubov-de Gennes (BdG) mode equations, and are normalized by $\int dx\left(|u_j(x)|^2 - |v_j(x)|^2\right) = 1$, and $\beta_j$ is a complex random variable with probability distribution

$$P(\beta_j) = \frac{2}{\pi}\exp(-2|\beta_j|^2). \tag{11}$$

We solve the BdG equation numerically, at time $t = 0$ before the step potential is swept, for the first 200 modes. The modes above this cutoff do not have an important dynamical effect on the condensate, and are omitted. This gives rise to the total number of excited atoms

$$N_{ex} = \sum_j(|\beta_j|^2 - \tfrac{1}{2})\int dx\left(|u_j(x)|^2 + |v_j(x)|^2\right) + \sum_j\int dx|v_j(x)|^2. \tag{12}$$

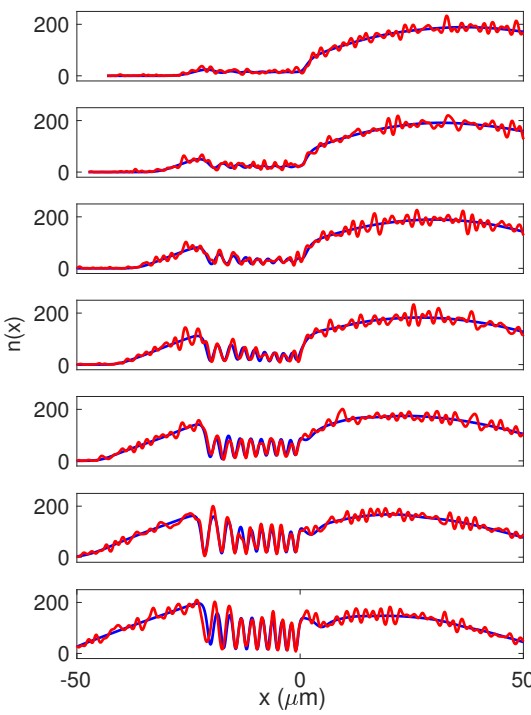

Figure 10: Comparison of condensate evolution with and without quantum fluctuations. Blue: GP simulation; red: a simulation with quantum noise obtained from the TWA method. The blue curves are the same as those in Fig. 8.

The last term corresponds to the quantum depletion, $N_{ex}^0$, with respect to a given condensate with $N_c^0$ atoms. If the total atom number ($N = N_c^0 + N_{ex}^0$) is held fixed, then the population in the condensate is given by $N_c = N - N_{ex}$. Since $N_{ex}$ fluctuates (with mean value $N_{ex}^0$) over individual realizations, $N_c$ then also fluctuates (with mean value $N_c^0$). The resulting stochastic wavefunction at time $t = 0$ is given by

$$\Psi(x) = \beta_0 \Psi_0(x) + \sum_j \left( \beta_j u_j(x) + \beta_j^* v_j^*(x) \right), \tag{13}$$

where $\beta_0 = \sqrt{N_c + 1/2}$ is the coefficient of the condensate mode.

The expectation value implied by (11) is $\langle |\beta_j|^2 \rangle = 1/2$, so $\langle N_{ex} \rangle$ is just the last term in (12), the quantum depletion of the condensate $N_{ex}^0$. The integral in the last term goes to zero as the wavelength drops below the healing length; here we find that for the condensate considered here it goes to zero for $j > 60$, and the quantum depletion converges to $\approx 34$ (which is about $0.006N$, where $N = 6000$). In the TWA simulation, the mean value of the excited atom number is $\langle N_{ex} \rangle \approx 33$, which is consistent with the quantum depletion, and its standard deviation is $\Delta N_{ex} \approx 13$.

Figure 10 shows one realization of the TWA simulation, compared with the GP simulation in Fig. 8. The spatial noise seen in the TWA realization arises from the quantum fluctuations. The standing wave features in the two simulations match closely with each other. The presence of the noise only leads to slight modulation of the standing wave amplitude.

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
