# Peer review of "Induced density correlations in a sonic black hole condensate"

_SciPost Physics, doi:SciPost Phys. 3, 022 (2017)_

## Round 2 · Referee Report · Anonymous · 2017-8-21

Strengths

1. Timely topic
2. The paper provides new insight into the role of correlations in the presence of sonic horizons

Weaknesses

none really

Report

In summary, this is a well written paper, which discusses the timely topic of correlations in the presence of analog event horizons. The paper seems technically correct, and importantly points out that in a recent experiment with a BEC where analog sonic horizons were identified, the density-density correlations may be attributed to atom number fluctuations rather than the quantum fluctuations. I enjoyed reading the paper. The authors carefully explain their methods, and in particular Section III C gives a clear picture of what is going on. To conclude, I do not have anything to complain about. It is a very good paper which will no doubt be well received by the cold atom community and those interested in analog gravity. I suggest the paper is published in its present form.

Requested changes

none

---

## Editorial Decision

published